# Genome-Wide Genetic Diversity and Population Structure of Tunisian Durum Wheat Landraces Based on DArTseq Technology

**DOI:** 10.3390/ijms20061352

**Published:** 2019-03-18

**Authors:** Cyrine Robbana, Zakaria Kehel, M’barek Ben Naceur, Carolina Sansaloni, Filippo Bassi, Ahmed Amri

**Affiliations:** 1Faculté des Sciences de Bizerte, Jarzouna, 7021 Bizerte, Tunisia; 2Banque Nationale de Gènes, Boulevard du Leader Yasser Arafat Z. I Charguia 1, 1080 Tunis, Tunisia; nour3alanour@yahoo.com; 3International Center for Agricultural Research in the Dry Areas (ICARDA), 10112 Rabat, Morocco; Z.Kehel@cgiar.org (Z.K.); F.Bassi@cgiar.org (F.B.); A.Amri@cgiar.org (A.A.); 4Centro Internacional de Mejoramiento de Maíz y Trigo (CIMMYT), El Batán, 56237 Texcoco, Mexico; C.Sansaloni@cgiar.org

**Keywords:** durum wheat, Tunisian landraces, center of diversity, genetic diversity, population structure, DArTseq technology

## Abstract

Tunisia, being part of the secondary center of diversity for durum wheat, has rich unexploited landraces that are being continuously lost and replaced by high yielding modern cultivars. This study aimed to investigate the genetic diversity and population structure of 196 durum wheat lines issued from landraces collected from Tunisia using Diversity Array Technology sequencing (DArTseq) and to understand possible ways of introduction in comparing them to landraces from surrounding countries. A total of 16,148 polymorphic DArTseq markers covering equally the A and B genomes were effective to assess the genetic diversity and to classify the accessions. Cluster analysis and discriminant analysis of principal components (DAPC) allowed us to distinguish five distinct groups that matched well with the farmer’s variety nomenclature. Interestingly, Mahmoudi and Biskri landraces constitute the same gene pool while Jenah Zarzoura constitutes a completely different group. Analysis of molecular variance (AMOVA) showed that the genetic variation was among rather than within the landraces. DAPC analysis of the Tunisian, Mediterranean and West Asian landraces confirmed our previous population structure and showed a genetic similarity between the Tunisian and the North African landraces with the exception of Jenah Zarzoura being the most distant. The genomic characterization of the Tunisian collection will enhance their conservation and sustainable use.

## 1. Introduction

Durum wheat is the tenth most important crop worldwide, grown on 13 M ha with a production of 39.1 Mt in 2017 (International Grain Council, Grain Market reports, 2017) and is mainly used for pasta production. It is mainly grown in the countries around the Mediterranean basin and its cultivation has extended to India, Mexico, North America, and Russia [1,2]. North Africa grows around 2.5 M ha of durum wheat with around 1 M ha in Tunisia. Landraces are still grown under traditional farming systems and are highly appreciated for local dishes such as frikeh, burghul, couscous, but their acreage is reduced significantly with the large adoption of new cultivars released since the 1970s [3,4].

Durum wheat (*Triticum turgidum* ssp. *durum*) is a self-pollinated allotetraploid cereal (2n = 4x = 28, AABB) originating from a cross between *Aegilops speltoides* (SS) and *Triticum urartu* (AA) and domesticated from primitive wheat (*Triticum turgidum* ssp. *dicoccum*) in the Fertile Crescent around 8000 years ago [5,6,7,8,9]. North Africa and Abyssinian regions are considered as secondary centers of diversity for durum wheat [10]. Tunisia, being part of the secondary center of diversity for durum wheat, has a rich diversity in terms of landraces and wild relatives [11,12].

Durum wheat landraces are valuable parental germplasm for wheat improvement programs around the world. Genesys database (www.genesys-pgr.com) contains 60,488 durum wheat accessions of which 22,600 are landraces. The International Center for Agricultural Research in the Dry Areas (ICARDA) genebank holds one of the largest collections of durum wheat accounting more than 20,531 accessions of which more than 65% are landraces. The Tunisian genebank (NGBT) holds a total of 4000 accessions of durum wheat most of which are pure lines collected from the predominant remaining landraces.

The Tunisian landraces are well adapted to a broad range of environments, genetically diverse, and are considered as an important reservoir of useful genes that could be exploited in wheat breeding programs [3,13,14]. A high genetic diversity of local Tunisian durum wheat has been reported using morphological, agronomic, physiological, and biochemical traits [4,5,6,7,8,9,10,11,12,13,14,15]. Sourour et al. [16] has shown important phenotypic diversity of the Tunisian germplasm estimated to 0.77 using Shannon–Weaver indices. Several molecular markers techniques were used to assess the genetic diversity in wheat including RFLP, RAPD, ISSR, AFLP, EST–SSR [17], DArT, and SNP [18,19]. A significant genetic difference using SSR markers was detected between landraces of durum wheat originated from Morocco and Syria using the Bayesian method and the Eigen analysis [20]. Medini et al. [21] reported high genetic diversity among 33 old Tunisian cultivars using 15 SSR markers. More recently, progress has been made for wheat in genomic research and genetic diversity analysis methods through the construction of a high-density SNP-based consensus map for durum wheat and the development of several next-generation sequencing (NGS) platforms [22,23]. An Axiom 35K array used to genotype 370 entries of durum wheat panel allowed us to differentiate among improved varieties and landraces and to show that Middle East and Ethiopia had the lowest level of allelic diversity compared to other regions [24]. Many studies have shown that genotyping-by-sequencing (GBS) has been increasingly adopted as a rapid and low-cost molecular technique for whole-genome SNP coverage [25], SNP discovery, genotyping, and genetic variability analysis for various crop species including durum wheat landraces [26,27,28]. The development of DArTseq™ technology has been applied successfully for large genomes such as barley [29] and polyploid or/and complex genomes such as tetraploid and hexaploid wheat [18,29,30]. The power of the DArTseq™ approach based on Illumina short read sequencing have been proven to be effective when used to study the genetic diversities of Syrian and Turkish durum wheat landraces [31,32], bread wheat [33], watermelon, and common bean landraces [34,35].

The present study aimed at (1) characterizing for the first time the population structure and the genetic diversity within and among the Tunisian durum wheat collection, (2) looking at potential mis-classification by linking off types of landraces to local modern durum wheat varieties and/or other landraces, and (3) comparing the Tunisian durum wheat landraces to landraces from the countries around the Mediterranean basin and West Asia region using high throughput DArTseq™ technology.

## 2. Results

### 2.1. DArTseq Marker Characteristics

In this study, a total of 110,856 DArTseq markers were identified in a set of 196 lines issued from six Tunisian durum wheat landraces, of which 16,148 markers were found to be of high quality and polymorphic, after removing markers with more than 20% missing data and with less than 5% minor allele frequency (MAF). Around 10% of the markers had 0 heterozygote alleles. The average of the polymorphism information content (PIC) value of the DArTseq markers was 0.165, and the median was 0.105. The distribution of the majority of MAFs was between 0.05 and 0.15 (Figure 1).

The DArTseq markers are well distributed across all the 14 chromosomes of durum wheat genomes based on the consensus bread wheat genetic map obtained from the International Wheat Genome Sequencing Consortium database (https://urgi.versailles.inra.fr/download/iwgsc/). The distribution is almost equal between A (4201 markers) and B (4659 markers) genomes with the highest number of markers observed on the chromosomes 7A and 2B. However, 7324 markers are still unassigned to any of the chromosomes (Table 1). A similar range of genetic diversity values (*H*_e_) is observed for both A and B genomes with an average of around 0.25. The maximum value of the expected heterozygosity (*H*_e_) was observed at chromosome 6B (0.28) and the minimum value was seen for chromosomes 3A, 4A, 5A, and 5B (0.24). For all chromosomes, the expected heterozygosity values (*H*_e_) were higher than the observed heterozygosity values (*H*_0_).

### 2.2. Genetic Distance and Clustering of the Tunisian Landraces

The allele sharing distance (ASD) among the 196 Tunisian landraces lines ranged from 0 to 0.79 with an average of 0.46 (Figure 2). However, different distance patterns were observed for different landraces. Jenah Zarzoura shows the lowest distance with an average of 0.1 and the lowest variability among its lines ranging from 0 to 0.16. For the Bidi landrace, the average distance is 0.17 with a range of 0.02–0.48. For Rommani, the average distance is 0.25 with values ranging between 0.02 and 0.63 and for Mahmoudi the average distance is 0.30 with a range of 0.35–0.58. Both Biskri and Jenah Khotifa show the highest average distances respectively of 0.43 and 0.44 with the largest variabilities reaching 0.78 and 0.61 respectively.

Cluster analysis of the 196 lines derived from the six Tunisian landraces was performed using the allele sharing distance (ASD) method and the results allowed us to group them into five clusters and to identify lines wrongly assigned to some landraces. The first cluster comprised all the lines of Jenah Zarzoura (Zar). The second cluster contained mainly Bidi lines (Bid). The third cluster is subdivided into two sub-groups, one containing Rommani lines (Rom) and the other having a mixture of lines from other landraces (seven from Biskri (Bis), three from Jenah Khotifa (jkf), and two from Bidi (Bid)). Cluster 4 had the majority of Jenah Khotifa lines (jkf) but contained also four lines of Rommani, two of Biskri, and one of Mahmoudi. The last cluster constitutes the largest one and gathered almost all lines of Biskri (Bis) and Mahmoudi (Mah) (Figure 3).

### 2.3. Population Structure of the Tunisian Landraces

The DAPC (discriminant analysis of principal components) results showed that 88 Principal Components (PCs) explain 82% of the total molecular variance. The optimum number of clusters was obtained with K = 6 using the Bayesian Information Criterion (BIC), which divided the lines into six sub-populations (Figure 4). The first subdivision level using the hierarchical population structure K = 2 separated Jenah Zarzoura lines (Zar) clearly from all the lines of the other landraces (Figure 5A). When K = 3, in addition to the separate sub-population of Jenah Zarzoura, two other sub-populations were identified, one combining most Mahmoudi (Mah) and Biskri (Bis) lines and the other included lines of landraces Bidi (Bid), Jenah Khotifa (jkf), and Rommani (Rom) (Figure 5B). With K = 4, a separate sub-population containing Jenah Khotifa lines appeared (Figure 5C); when K = 5, all the landraces were assigned to different sub-populations, except for Biskri and Mahmoudi, which remained grouped in the same sub-population (Figure 5D). The number of sub-population K = 5 was then chosen to differentiate between different landraces and the sub-populations were given the following names: BID for Bidi, BIS+MAH for Biskri and Mahmoudi, JKF for Jenah Khotifa, ROM for Rommani, and ZAR for Jenah Zarzoura.

Increasing K to 6 resulted into the formation of an additional small group composed by 13 lines mainly of Biskri (7) and Jenah Khotifa (4) showing that these lines could be off-types within their respective landraces, which need further characterization (Figure 5E). The global population structure image was maintained for the values of K equal to 7 and 8 with all different landraces assigned to different sub-populations except the landraces Biskri and Mahmoudi that are still grouped in the same sub-population (Figure 5F).

The results of analysis of molecular variance (AMOVA) for K = 2 to K = 7 showed that the variance between populations was high for K = 2 (31.92%), and it decreased for K = 3 and K = 4 (28.75% and 31.29%, respectively) to increase and reach a plateau for K = 5 with a slightly higher value (31.77%). The variance between lines within populations decreased from 11.32% for K = 2 to reach values close to 0% for K = 5. Comparing AMOVA results between populations with K = 5 and the real groups using the landrace’s name clearly support the hypothesis that K = 5 better differentiates genetically the lines than the characterization of landraces by name, resulting in higher variance between populations (31.77% for K = 5 and 28.79% for real groups), lower variance between lines 68.23% for K = 5 and 70.39% for real groups, and lower variance between lines within populations (Table 2). Thus, these findings indicated a higher genetic variation among rather than within Tunisian landraces.

Results from hierarchical AMOVA using hierarchical subdivision strata from K = 2 up to K = 5 indicated that most of the molecular variance was explained with the first level K = 2 separating Jenah Zarzoura from the other landraces (17.18%) and the fifth level K = 5 within the four groups (26.62%) separating Bidi, Rommani, and Jenah Khotifa and the large group constituted by Biskri and Mahmoudi. The remaining molecular variance (variance between lines) was also low compared to non-hierarchical AMOVA results (60.55%) using K = 5 (Table 3).

Finally, the resulted variance components from AMOVA analysis with K = 5 were significant when they were tested using permutations (Figure 6). These results showed that the structure found by number of groups with K = 5 was valid and not a result of a random effect.

### 2.4. Genetic Diversity and Genetic Distance between Tunisian Landraces

The genetic distance between landrace populations showed that Jenah Zarzoura sub-population is the farthest from all the other landraces, from Rommani (0.820), from Biskri and Mahmoudi (0.730), from Jenah Khotifa (0.682), and from Bidi (0.816) (Table 4). Furthermore, the Jenah Khotifa sub-population was equally distant to the landraces Bidi and Rommani (0.557). These results confirmed the population subdivision found by DAPC, which separated the Jenah Zarzoura sub-population from the other landraces when K = 2 and grouped the sub-populations Jenah Khotifa, Bidi, and Rommani when K = 3.

Results of genetic diversity estimates in each sub-population obtained based on DAPC with K = 5 show that the highest genetic diversity was observed within the Jenah Zarzoura and Rommani populations (*H*_e_ = 0.27). The lowest genetic diversity was observed within the Bidi population (*H*_e_ = 0.12). Biskri and Mahmoudi forming the same group with 73 lines showed a moderate genetic diversity (*H*_e_ = 0.23). Based on *F*_ST_ values, Jenah Zarzoura showed a low *F*_ST_ value of 0.05. Thus, this population presents a genetic drift compared to the others, having a fixation of alternate alleles with *F*_ST_ values superior to 0.25 (Table 5). The results from genetic diversity confirmed the results obtained using the allele sharing distance (ASD) between lines within the same population as shown in Figure 1.

### 2.5. Linking the Mis-Classified Lines to Local Landraces and Improved Varieties or/and to the ICARDA/CIMMYT Elite Lines

Cluster analysis based on the ASD method was performed using a set of 27 Tunisian durum wheat landrace populations, six local improved varieties, and seven ICARDA/CIMMYT inbred elite lines, along with lines which formed the additional sub-population when K = 6 from the previous set. This study aimed at showing the relationships among a larger number of Tunisian landraces, the differences with improved varieties, and germplasm and at shedding more light on the mis-classified lines included in the last sub-population where K = 6.

Results of cluster analysis showed three main clusters (Figure 7). Cluster 1 contained most of the improved varieties released in Tunisia and the advanced lines from ICARDA and CIMMYT, along with three landraces; Azizi (Aziz P), Chetla 1(Chet1 P), and Jenah Khotifa 2 (JK2 P). The six mis-classified lines of Biskri (Bis) from the additional sub-population when K = 6 in the analysis of Set 1 are included in this cluster and are grouped with the improved variety Khiar (Khia T). Cluster 2 contained the majority of the Tunisian landraces along with two improved varieties: INRAT69 (INRA T) and Karim (Kari T). This cluster could be subdivided into two groups: a small group composed of six local landraces (Arbi (Arbi P), Rommani 3 (Rom3 P), Biskri 1(Bis1 P), Chetla 2 (Chet2 P), Richi (Richi P), and Agili (Agil P)) and the two improved varieties, Karim (Kar T) and INRAT 69 (INRA T) and a large group that can be further divided into two sub-groups, one of which constitutes a separate group with only Jenah Zarzoura lines (Zar1 L, Zar2 L, and Zar3 L) and the other of which is composed of the majority of local landraces: Jenah Khotifa 1(jkf1 P), Sbei (Sbei P), Mahmoudi (Mah P), Souri (Sour P), Bayadha (Baya P), Swabei Algia (SA P), Aouadi (Aoua P), Hamira (Hmir P), Rommani (Rom1 P), Bidi 1(Bid1 P), Bidi 3 ( Bid3 P), Rommani 2 (Rom2 P), Wardbled (WB P), Derbessi (Derb P), Chili (Chil P), Biskri 2 (Bis2 P), and Bidi 2 (Bid2 P). This last sub-group showed that the two lines of Jenah Khotifa (Jk2 L and Jk4 L) from Set 1 are clustered with Jenah Khotifa population 1(jkf1 P). The Rommani line (Rom1 L) is grouped with the Rommani population (Rom1 P) and the line Bidi 25 (Bid25 L) is grouped with Bidi populations (Bid1 P and Bid3 P). Cluster 3 was divided into two groups: the first group contained a mixture of lines including Jenah Khotifa 11 line (Jk11 L), Biskri 12 line (Bis12 L), and Mahmoudi 8 line (Mah8 L) and the second group showed that the mis-classified lines of Jenah Khotifa 19 (Jk19 L) and Mahmoudi 30 (Mah30 L) included in the additional sub-population when K = 6 of Set 1 analysis are clustered with the ICARDA/CIMMYT inbred line (MCHCB-102). These results showed that most of the lines included in the mixed sub-population of the DAPC analysis with K = 6 are grouped with improved varieties and germplasm and showed that some landraces having the same local name are classified in different clusters.

### 2.6. Comparison of Tunisian Landraces to Landraces from Mediterranean and West Asia Regions

The population structure of the Mediterranean and West Asian landraces along with four lines representing each of six Tunisian landraces included in Set 1 was assessed using the DAPC method. The optimum number of groupings is determined with K = 12 based on the BIC criterion using 66 PCs, which explained 82% of the total molecular variation. Sub-population 6 showed that the Tunisian landraces Rommani (Rom), Jenah Khotifa (jkf), and Bidi (Bid) were grouped with the majority of the Algerian landraces (8); Sub-population 12 included all Ethiopian landraces (11), along with one accession from Afghanistan and two accessions from Yemen; Sub-population 9 included the Tunisian landraces Biskri (Bis) and Mahmoudi (Mah) that were grouped with the remaining five landraces from Tunisia, with the majority of Algerian and Lebanese landraces (6), and with some Moroccan landraces (3). Sub-population 11 included exclusively the Tunisian landrace Jenah Zarzoura (Zar) with two accessions from Jordan (Figure 8). The population structure results confirmed that the newly collected landrace Jenah Zarzoura constitutes a new gene pool and that the Tunisian landraces are genetically closer to North African landraces.

## 3. Discussion

The molecular markers techniques are important tools for better understanding genetic diversity, undertaking association mapping and ensuring efficient conservation and management of genetic resources. This study demonstrated the relevance of DArTseq technology as a reliable and cost-effective tool for assessing the diversity within and between landraces and for comparing them with other germplasm of durum wheat. This technique yielded a large number of polymorphic and informative markers equally spread in the A and B genomes allowing high coverage of the genomes of the Tunisian durum wheat germplasm compared to other molecular techniques used previously. Similar genomes coverage was found on a panel of 170 durum wheat entries [27]. The large coverage of the genomes can serve to undertake association mapping in the studied germplasm, including finding new allelic variations for major breeders sought traits such as QTLs found by other studies in chromosomes 7A and 2B linked to protein content [36], gluten strength and yellow pigment [37,38], and salinity tolerance and yield components [39,40]. DArTseq technology along with other high throughput and genotyping by sequencing molecular techniques are increasingly used to study the genetic diversity of different crops as they allow to study the genetic diversity of large number entries and complex genomes [41,42].

### 3.1. DArTseq Polymorphism of Tunisian Durum Wheat Landraces

Polymorphic information content (PIC) values revealed by DArTseq markers averaged 0.165 with an asymmetric distribution skewed towards low values. The same distribution tendency was found using the same approach for 91 durum wheat landraces from the Fertile Crescent and for 138 wheat germplasm from Southwest China [32,41]. Ren et al. [43] have shown the same PIC value for North African durum accessions (0.168) using 946 SNP markers as part of a genetic diversity study in a worldwide durum wheat germplasm collection. Recent studies using DArTseq markers reported higher PIC values for worldwide durum wheat accessions (0.35) and for accessions originating from central Fertile Crescent (0.26) [27,32]. A previous study on 34 Tunisian durum wheat old varieties reported a PIC value of 0.68 [25]. This difference is explained by the bi-allelic nature of DArT markers for which the maximum value for PIC is 0.5 compared to multi-allelic SSR markers with the maximum PIC value of 1 [44].

The lines of the six Tunisian durum wheat landraces showed a moderate level of genetic diversity (*H*_e_ = 0.25), which is higher than the one exhibited by a set of world-wide durum wheat collection (*H*_e_ = 0.224) using SNP markers [43]. Our results confirmed previous findings showing that the Fertile Crescent and the eastern Mediterranean durum landraces are more diverse than those from the Western Mediterranean regions [45,46]. Although several studies revealed high genetic diversity in the Tunisian landraces based on phenotypic characterization [15,16], the moderate level of genetic diversity could be explained by the fact that the Tunisian durum wheat lines included in this study are derived from six landraces collected from a limited geographic area in the center and the south parts of Tunisia.

### 3.2. Genetic Diversity and Population Structure of the Tunisian Durum Wheat Landraces

Allele sharing distance using DArTseq markers allowed us to differentiate among the six landraces Jenah Zarzoura, Biskri, Jenah Khotifa, Mahmoudi, Bidi, and Rommani by showing different distance patterns. Two methods, clustering analysis based on ASD (allele sharing distance) and DAPC (discriminant analysis of principal components) were used for depicting the genetic relationship and structure of the Tunisian collection. The first method classified the panel into five groups matching mostly with the farmer’s landraces names, with Jenah Zarzoura being the most distant and Mahmoudi and Biskri included in the same cluster. The closeness of Mahmoudi and Biskri is for a long time reported by Boeuf [11] based on glume and spike color, which could be due to the exchange of these landraces among farmers from Algeria and Tunisia [47]. The clear distinction of Jenah Zarzoura could be due to its confinement to a geographically limited area of the Mareth oasis or to a different pattern of introduction.

The use of the multivariate method DAPC to evaluate the population structure showed better performance and allowed for a population subdivision similar to other studies [24,48]. Several molecular approaches were used to assess the population structure in durum wheat landraces from SSRs to DArTs [49,50]. More recently, GBS-SNPs and DArTseq approaches were mainly used for hexaploid wheat population structure studies [28,51], and some reports have described a durum wheat population structure based on the DArTseq technique that allowed to classify the Turkish and Syrian durum wheat landraces in the same gene pool [32]. The DAPC analysis results were in concordance with those of clustering analysis, despite minor differences. Both showed a good fit between the grouping and the names of the varieties, which reflects the ability of farmers to differentiate among the landraces. However, a small group composed of a mixture of landraces appeared, and some lines of the differentiated landraces are included in different clusters, which could be due mainly to mis-naming of the landraces during the collecting missions and to possible mixtures in the landraces. These findings confirm the multiline nature of landraces that is also found through morphological and agronomic characterization [52]. This heterogeneity offers an important buffering capacity of landraces in drought-prone and fluctuating environments.

Hierarchical AMOVA analysis based on hierarchical subdivision strata from K = 2 up to K = 5 agreed with DAPC analysis results and supported the high level of molecular variance to K = 2 (17.18%) and to K = 5 (26.62%). AMOVA analysis results on the basis of the landraces’ names indicated a higher genetic variation among (28.79%) rather than within (0.82%) Tunisian landraces. Taking in consideration the structure of the population based on the optimal number of grouping K = 5, AMOVA results showed an increase in percent of the explained genetic variance among landraces (~31.77%) and a decrease of the genetic variance within them. Soriano et al. [49] revealed much variability within sub-populations (83%) than between them (17%) and Mangini et al. [53] found higher genetic diversity within the two Italian durum wheat landraces ‘‘Bianchetta’’ and ‘‘Grano Ricco’’ (9.5 and 9.4%, respectively) and low genetic diversity within ‘‘Dauno III.”.

This study allowed us to assess for the first time the genetic diversity within and among the Tunisian durum wheat landraces using the DArTseq technique, which allowed us to show different levels of genetic diversity within landraces. Jenah Zarzoura and Rommani populations showed the highest genetic diversity (*H*_e_ = 0.27), and the Bidi landrace showed the lowest genetic diversity (*H*_e_ = 0.12). The low genetic variation within the landraces could be explained by the selection from farmers for desirable traits and/or from environmental conditions pressure. Compared to the other landraces, the Jenah Zarzoura landrace showed a high expected heterozygosity (*H*_e_ = 0.268) and a low fixation index (*F*_ST_ = 0.05). Thus, this small differentiation could be explained by the confinement of this landrace to a specific environment in the oasis of Mareth, which reflects the geographic isolation of the oasis of Mareth, limited seed exchange, and selection pressure by farmers. Most often, farmers are selecting the best representative spikes from a landrace to form the seed lots.

### 3.3. Origin of Tunisian Durum Wheat Landraces

The cluster analysis extended to 27 other Tunisian landraces, improved varieties, and germplasm along with the lines included in the mixed cluster when K = 6 from Set 1 showed large genetic diversity among the germplasm studied. The six lines of landrace Biskri mis-classified in the additional sub-population when K = 6 in the Set 1 analysis were grouped with the modern cultivar Khiar, and the two mis-classified lines of Mahmoudi and Jenah Khotifa were grouped with the elite germplasm (MCHCB-102). These results confirm the possibility of mixture in some landraces, which could be due to seed exchange and threshing practices as suggested by other studies [52,54]. This extended genetic study confirms the uniqueness of the Jenah Zarzoura landrace and the classification of Jenah Khotifa, Rommani, and Bidi lines with their respective landraces, but not with the other populations with the same name. This could be due to the mis-naming of landraces during seed exchange among farmers or during the collecting missions undertaken by the genebank teams.

Tunisia is considered as a secondary center of diversity for durum wheat, and the introduction patterns of durum wheat landraces into Tunisia and North Africa are still under discussion. The DAPC analysis including landraces from Tunisia and landraces from the countries around the Mediterranean and West Asia regions allowed us to define 12 distinct groups, which can be used to highlight the relationships between Tunisian landraces and other landraces. Most Tunisian landraces held at the ICARDA genebank as well as the lines derived from landraces collected recently in Tunisia were grouped with landraces from North Africa neighboring countries and with landraces from Greece, Italy, and Lebanon. Jenah Zarzoura remained distinct from all landraces studied except for the two landraces from Jordan. These results suggest that most Tunisian landraces could be obtained through Lebanon via Greece and Italy, while Jenah Zarzoura was obtained through another introduction pathway. Previous reports have demonstrated two dispersal patterns of durum wheat in the Mediterranean Basin from the Fertile Crescent: over the north side via Turkey, Greece, and Italy and the south side via North Africa [55]. Moragues et al. [56] supported this hypothesis and classified a collection of durum wheat landraces originating from different Mediterranean countries in two groups: (i) landraces from the North and East Mediterranean basin and (ii) landraces from North Africa and the Iberian Peninsula. More recently, Soriano et al. [49] showed an eastern–western dispersal of the Mediterranean durum landraces, which have been classified into four sub-populations: (i) Eastern Mediterranean, (ii) the Eastern Balkans and Turkey, (iii) the Western Balkans, and (iv) Egypt and the Western Mediterranean. The grouping of landraces from North Africa with Italy and Greece was also confirmed by Olivera et al. [57] and Nazco et al. [58], and could be explained by the Roman influence on durum wheat cultivation in North Africa. The early development of Carthage trade maritime activity in the Mediterranean sea enhanced seed exchanges between Tunisia and the Mediterranean countries [59], which might explain the grouping of the Bidi, Jenah Khoutifa, and Rommani lines with the majority of Algerian landraces (Sub-population 6) and that of the Biskri and Mahmoudi lines with Lebanese and Moroccan landraces (Sub-population 9). Moreover, our work confirmed that Biskri and Mahmoudi lines constitute the same gene pool and that Jenah Zarzoura lines constitute a new gene pool distant from the other Tunisian and foreign landraces, which were grouped with only two accessions from Jordan (Sub-population 11). A possible explanation is that the Jenah Zarzoura population, which was collected from the oasis of Mareth, located in the south of Tunisia, near the Mediterranean Sea, might have been introduced from the east through different paths, probably from Egypt to neighboring countries and possibly received from Palestine and Jordan [60,61], or through the introduction by the Phoenicians coming from Lebanon to Carthage between the 9th and 2nd centuries B.C [59]. During the Roman period (7th to 3rd centuries B.C.), Tunisia became the breadbasket of the Italian peninsula and the source of the excellent semolina quality from durum wheat grown in North-African countries [62]. The landraces of Tunisia and North Africa have also be traced to the introductions by Romans, who contributed to the modernization of the irrigation systems and extended wheat cultivation to Southern Tunisia [59].

### 3.4. Implications on Conservation and Use of Genetic Resources

The results of this study can be used to direct future activities of collection, conservation, and the use of durum wheat genetic resources in Tunisia. In terms of adding new diversity to the existing collections, more collection is needed in Tunisia mainly in the oasis areas to collect different landraces like Jenah Zarzoura. Future studies of this kind of germplasm will shed more light on the specific nature of this germplasm as in the case of durum wheat germplasm from Ethiopia [24]. Additionally, more landraces from other regions, even if they have the same local names, should be collected and given different identifiers within the genebank database and considered as different accessions in the ex situ collection. When collecting, the team should avoid plants with characteristics of improved varieties to avoid mixtures. For ex situ conservation, the genebank in Tunisia should conserve the bulk seeds of each landrace instead of seeds of many individual lines constituting each landrace. This will reduce the cost of conservation and avoid conserving several copies of the same line. DArTseq markers have also allowed us to identify outlier lines within the landraces and can therefore be eliminated during multiplication and characterization. For landraces still prevailing under traditional farming systems and under harsh conditions, on-farm conservation could be promoted to conserve a larger genetic base and the associated local knowledge. The need for ensuring long-term conservation of Tunisian durum wheat landraces is dictated by the on-going genetic erosion due to the spread of newly released cultivars and by their richness in genes for tolerance to drought, heat, and salinity and their quality attributes for different end uses [63].

## 4. Materials and Methods

### 4.1. Plant Material

This study used three sets of germplasm:Set 1: A total of 196 pure lines issued from six Tunisian durum wheat landraces known as Mahmoudi, Rommani, Jenah Zarzoura, Bidi, Jenah Khotifa, and Biskri, collected from 5 regions between 2009 and 2010 and conserved by the National Genebank of Tunisia (NGBT), were used for assessing the intra and inter genetic diversity (Table 6). The landraces Mahmoudi and Jenah Zarzoura are constituted by 30 lines each, and Rommani and Bidi are constituted by 33 lines each. The landraces Biskri and Jenah Khotifa are represented by two populations each: Biskri1 (31), Biskri2 (7), Jenah Khotifa1 (29), and Jenah Khotifa2 (3).Set 2: A total of 40 accessions composed of six improved varieties released in Tunisia, seven ICARDA/CIMMYT elite inbred lines and 27 Tunisian durum wheat landraces (15 landraces are represented by one accession; Jenah Khotifa, Biskri, and Chetla are represented by two accessions each; Bidi and Rommani are represented by three accessions each) were used for identification of potential mis-classified lines from Set 1 and for comparison between Tunisian durum wheat landraces and improved germplasm (Table 7).Set 3: A total of 207 durum wheat landraces collected from Mediterranean and West Asia countries—Morocco (17), Algeria (14), Tunisia (13), Libya (9), Egypt (12), Lebanon (10), Syria (11), Jordan (11), Israel/Palestine (10), Iraq (11), Iran (12), Afghanistan (16), Yemen (10), Greece (18), Cyprus (10), Italy (12), and Ethiopia (11)—randomly chosen from the ICARDA genebank collection along with four lines representing each Tunisian landraces from Set 1 were used for a genetic relationship study ( Table 8).

The seeds for Sets 1 and 2 were taken from single plant spikes of each line, landrace, and germplasm grown at the Mornag INRA-Tunisia experiment station during the 2014–2015 growing season, while the seeds for Set 3 were send to CIMMYT from the ICARDA genebank within the joint effort to genotype wheat genetic resources.

### 4.2. Genotypic Characterization Using the DArtseq™ Method

Fresh young leaves from a single individual plant per accession have been used for genomic DNA extraction performed through a modified CTAB (cetyltrimethylammonium bromide) method [64]. The DNA quality was determined by electrophoresis using 1% agarose gel and quantified with NanoDrop 8000 spectrophotometer V 2.1.0.

A high-throughput genotyping method using DArTseq^TM^ technology was employed to generate a genomic profile of the germplasm at the Genetic Analysis Service for Agriculture (SAGA) facility at CIMMYT, Mexico. This method used a combination of two restriction enzymes (PstI and HpaII) to reduce the genome complexity and to generate a genomic representation of the samples [31]. The genomic DNA was submitted to a process of digestion and ligation with a specific PstI-RE site-adapter tagged with 96 different barcodes, which allow to multiplex 96 DNA samples in a single lane of Illumina HiSeq2500 instrument (Illumina Inc., San Diego, CA, USA). The amplified fragments were sequenced up to 77 bases, generating around 500,000 unique reads per sample. A FASTQ files (full reads of 77 bp) were quality filtered using a Phred quality score of 30, which represents a 90% base call accuracy for at least 50% of the bases. An additional filter has been applied on barcode sequences. DArTsoft 14 was used to generate Silico-DArT score tables as data (1/0), indicating the presence/absence variation (PAV) markers and SNP markers.

### 4.3. DArTseq Markers and Cluster Analysis

DArTseq markers were mapped using the consensus map version 4.0 (www.diversityarrays.com) developed by DArT Pty. Ltd., Australia, and the reference wheat genome issued from the International Wheat Genome Sequencing Consortium database (IWGSC WGAv0.4), available online at https://urgi.versailles.inra.fr/download/iwgsc/.

DArTseq raw data was filtered according to markers criterion; minor allele frequency >5% and missing data ≤20%.

The summary statistics of the filtered DArTseq markers such as the expected heterozygosity (He) or genetic diversity (GD), minor allele frequency (MAF), and the polymorphic information content (PIC) [44], were calculated using R-project (http://www.r-project.org/) [65].

For cluster analysis of the collection, allele sharing distance matrix was computed as described by Goa et al. [66]. The distance between individuals i and j was defined as
$$D_{\mathrm{ij}}=\frac{1}{L}\overset{L}{\underset{l=1}{\sum }}d_{\mathrm{ij}}\left(l\right)$$where L is the total number of markers, and dij(l) is 0, 1, or 2 if individuals i and j have zero, one, or two allele(s) in common at Locus l. Classification of the individuals into groups was performed using the allele sharing matrix and Ward’s minimum variance algorithm [67]. The clustering algorithm was run using the hclust function within the R package [68].

### 4.4. Population Structure and Genetic Differentiation

Discriminant analysis of principal components (DAPC) was used to infer the number of clusters of genetically related individuals [48], using the *adegenet* package and *popr* in R-project [68]. DAPC is a multivariate analysis requiring three steps; first the data is transformed using principal component analysis, sub-groups are then identified using k-mean clustering, and discrimination between the sub-groups is then optimized using discriminant analysis. For the k-mean clustering, the optimal number of groups was identified using the Bayesian information criterion (BIC) as a measure of goodness of fit. The number of sub-groups (K) was set from 2 to 8 and the *K*-value with the lowest BIC was retained as the optimal number of clusters. A discriminant analysis was then implemented using the groups found by k-mean clustering [69].

For detecting the genetic variation among and within population(s) and supporting the hypothesis of the population structure, analysis of molecular variance (AMOVA) was performed for different hierarchical subdivision levels as well as for the full population structure strata from K = 2 to K = 8. Significance levels for variance component were estimated based on 10,000 permutations using the *randtest* function in the R-project as described by Excoffier et al. [70].

For genetic differentiation and relationships among the sub-populations issued from the population structure analysis, the genetic distance between the sub-populations using Reynolds genetic distance was computed [71], and the genetic indices, such as the observed heterozygosity (*H*_o_, the proportion of loci that are heterozygote for a population), the expected heterozygosity or genetic diversity (*He*, the fraction of all landraces which would be heterozygote for any randomly chosen locus), and the F-statistics (*F***_ST_**) as developed by Wright [72], were calculated.

## Figures and Tables

**Figure 1 ijms-20-01352-f001:**
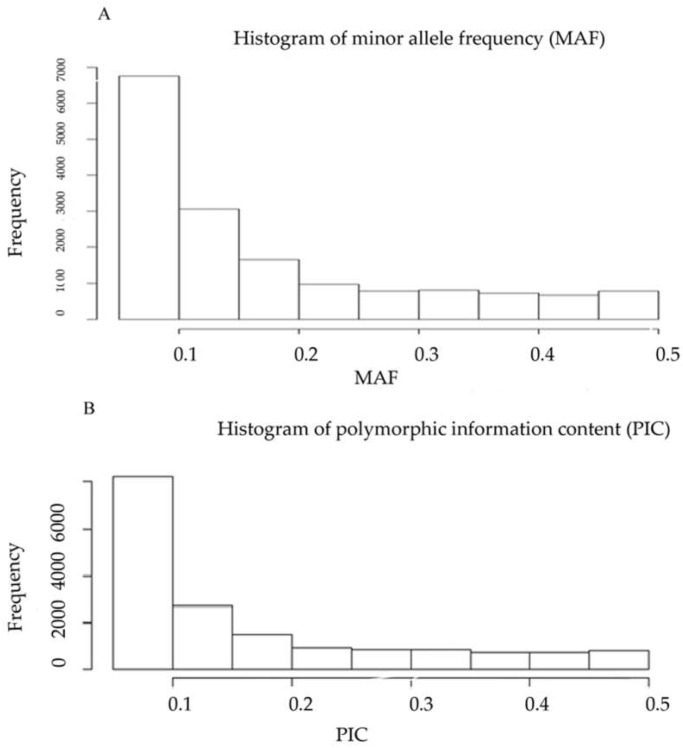
Frequency distribution of (**A**) minor allele frequency (MAF) and (**B**) polymorphic information content (PIC) of 16,148 DArtseq markers.

**Figure 2 ijms-20-01352-f002:**
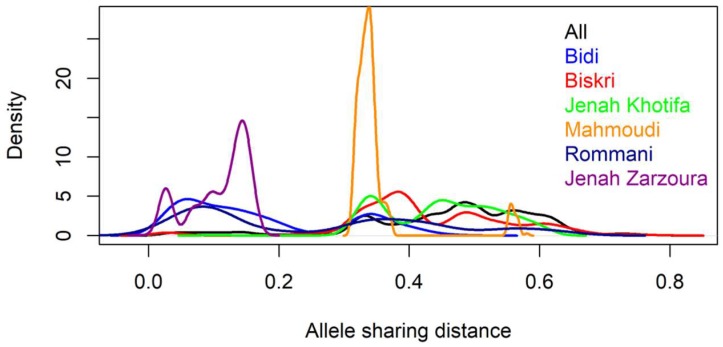
Distribution of allele sharing distance of 196 pure lines derived from six Tunisian durum wheat landraces using DArTseq markers.

**Figure 3 ijms-20-01352-f003:**
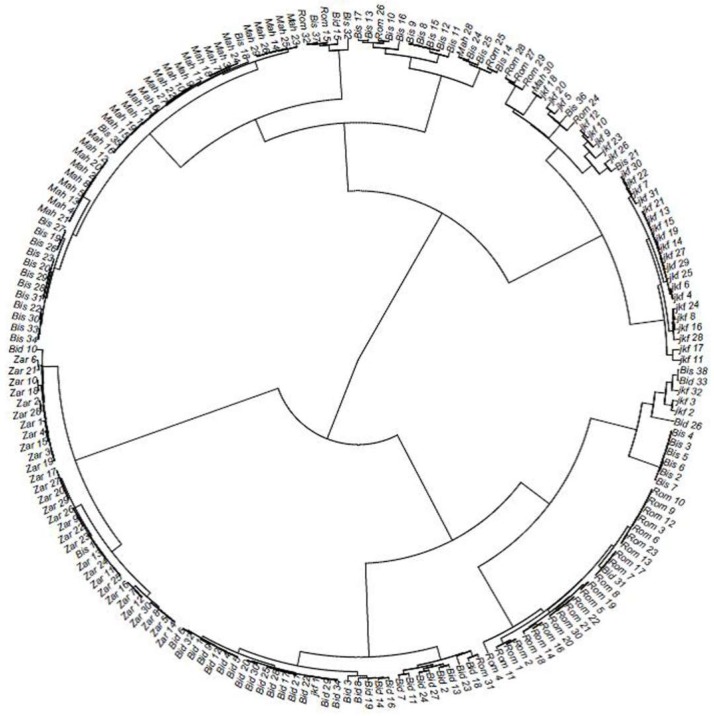
Cluster tree of 196 lines derived from six Tunisian durum wheat landraces based on allele sharing genetic distance. Bid: Bidi; Mah: Mahmoudi; Rom: Rommani; Bis: Biskri; jkf: Jenah Khotifa; Zar: Jenah Zarzoura.

**Figure 4 ijms-20-01352-f004:**
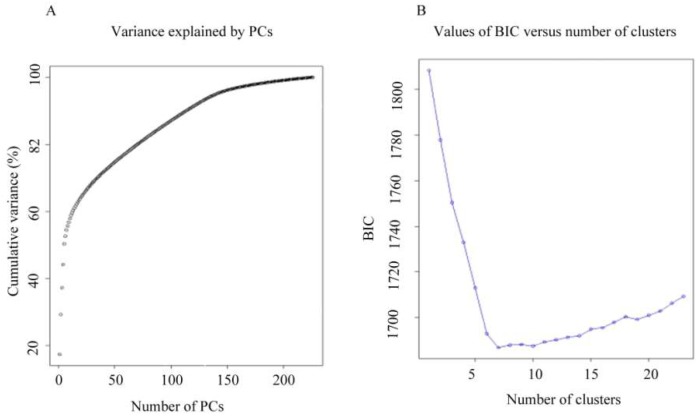
Optimal number of sub-populations using discriminant analysis of principal components (DAPC). (**A**) Cumulative variance explained by PCs; (**B**) BIC versus number of clusters.

**Figure 5 ijms-20-01352-f005:**
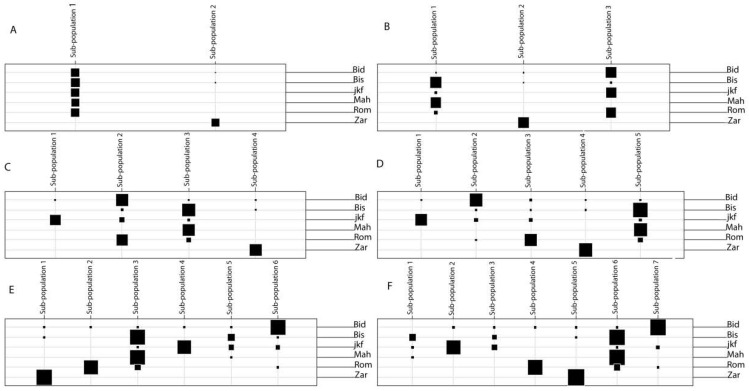
Table graphs comparing distribution of the original landrace classification to the sub-populations using DAPC with (**A**) K = 2; (**B**) K = 3; (**C**) K = 4; (**D**) K = 5; (**E**) K = 6; (**F**) K = 7. K: Number of sub-populations; Bid: Bidi; Mah: Mahmoudi; Rom: Rommani; Bis: Biskri; jkf: Jenah Khotifa; Zar: Jenah Zarzoura.

**Figure 6 ijms-20-01352-f006:**
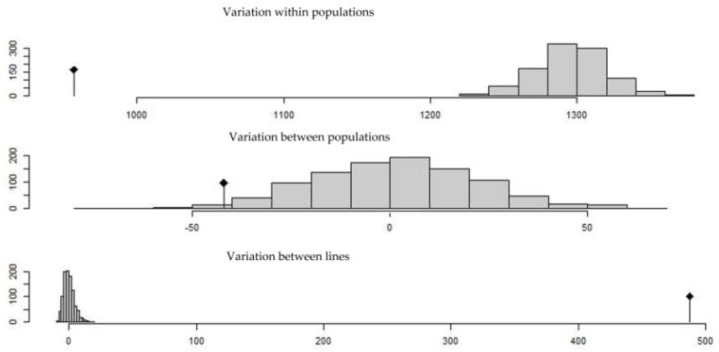
AMOVA permutations test using 1000 permutations for number of sub-population K = 5.

**Figure 7 ijms-20-01352-f007:**
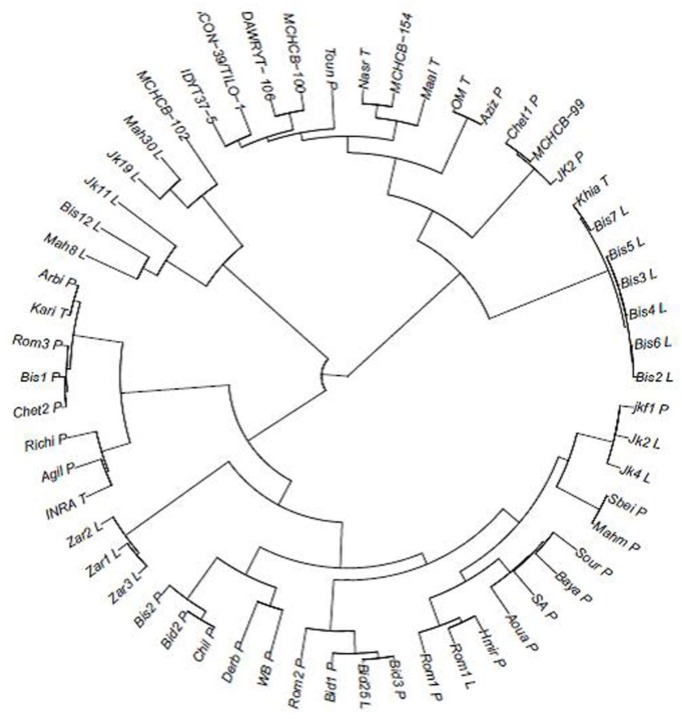
Dendrogram of Tunisian durum wheat landraces, local improved varieties, and ICARDA/CIMMYT elite lines based on allele sharing genetic distance. L: Line; P: Population. Local landraces: Bid: Bidi; Mah: Mahmoudi; Rom: Rommani; Bis: Biskri; jkf: Jenah Khotifa; Zar: Jenah Zarzoura; Aziz: Azizi; Chet: Chetla; Sbei; Sour: Souri; Baya: Bayadha; SA: Swabei Algia; Aoua: Aouadhi; Hmir: Hamira; WB: WardBled; Derb: Derbessi; Chil: Chili; Agil: Agili; Richi; Arbi; Toun: Tounsia. Local improved varieties: OM: OmRabii; Nasr; Maal: Maali; Khia: Khiar; INRAT: INRAT69; Kari: Karim. ICARDA/CIMMYT elites lines: MCHCB-102: ICARDA inbred line; MCHCB-100: IcaJoudy1; DAWRYT-106: Nachit; MCHCB-154: Zeina4; Con39/Tilo-1: Louiza; IDYT37-5: Ammar 6; MCHCB-99: Ammar 10.

**Figure 8 ijms-20-01352-f008:**
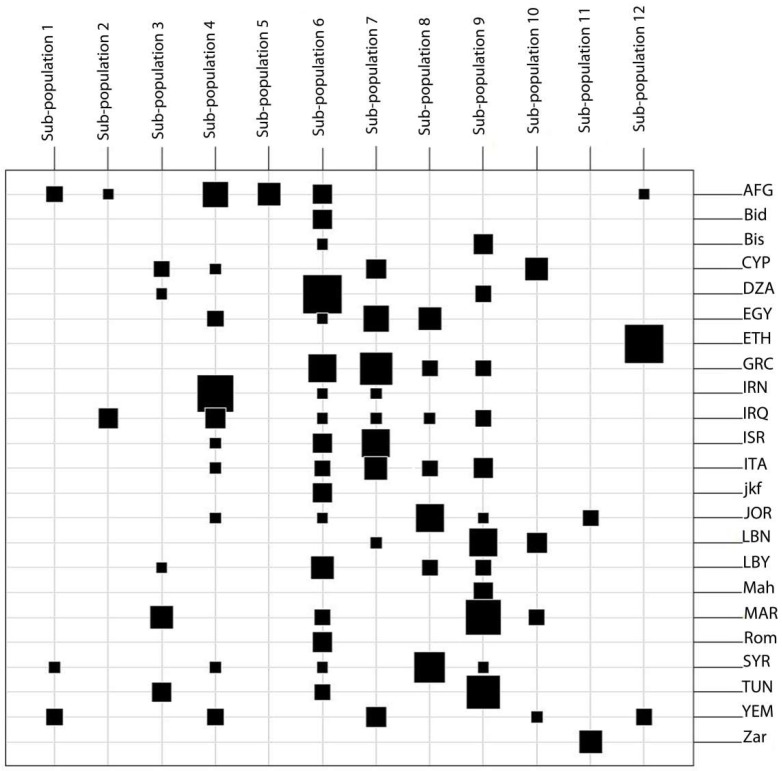
Table graphs comparing Mediterranean, West Asian, and Tunisian landraces classification to the sub-populations using DAPC with K = 12. AFG: Afghanistan; CYP: Cyprus; DZA: Algeria; EGY: Egypt; ETH: Ethiopia; GRC: Greece; IRN: Iran, IRQ: Iraq; ISR: Palestine/Israel; ITA: Italy; JOR: Jordan; LBN: Lebanon; LBY: Libya; MAR: Morocco; SYR: Syria; TUN: Tunisia; YEM: Yemen.

**Table 1 ijms-20-01352-t001:** Number of the selected DArTseq markers (*n*), the observed heterozygosity (*H*_0_), and the expected heterozygosity (*H*_e_) across the 14 chromosomes in 196 Tunisian durum wheat landraces lines based on the consensus bread wheat genetic map.

	A Genome			B Genome		
Chromosome	*n*	*H* _e_	*H* _o_	*n*	*H* _e_	*H* _o_
**1**	562	0.27	0.15	658	0.26	0.14
**2**	569	0.25	0.14	864	0.27	0.13
**3**	633	0.24	0.14	777	0.26	0.14
**4**	511	0.24	0.16	429	0.25	0.16
**5**	587	0.24	0.18	639	0.24	0.16
**6**	452	0.25	0.15	573	0.28	0.15
**7**	797	0.25	0.14	719	0.26	0.15
**Total**	4201			4659		
**Unassigned**	7324	0.25	0.18			

**Table 2 ijms-20-01352-t002:** AMOVA summary for comparison between different numbers of sub-populations K and the real groups of lines issued from six Tunisian durum wheat landraces.

Variance			
Number of Sub-Populations	Among Landraces (%)	Among Lines within Landrace (%)	Among Lines (%)
K = 2	31.92	11.32	56.76
K = 3	28.75	5.60	65.65
K = 4	31.29	1.59	67.12
K = 5	31.77	0.00	68.23
K = 6	31.71	0.00	68.29
K = 7	31.51	0.00	68.49
Real groups	28.79	0.82	70.39

**Table 3 ijms-20-01352-t003:** Hierarchical AMOVA results from K = 2 to K = 5 using hierarchical subdivision strata of 196 lines derived from the six durum wheat Tunisian landraces.

Subdivision Strata	Variance Components	Percentage
Variations between K = 2	271.63	17.18
Variations between K = 3 and K = 2	0	0
Variations between K = 4 and K = 3	13.46	0.85
Variations between K = 5 and K = 4	420.82	26.62
Variations between lines and K = 5	957.20	60.56
Variations between lines	0	0
Total variations	1580.63	100

**Table 4 ijms-20-01352-t004:** Reynolds genetic distance between Tunisian durum wheat landrace populations based on sub-populations number K = 5.

Landraces	BID	ZAR	BIS + MAH	JFK
**ZAR**	0.816			
**BIS+MAH**	0.624	0.730		
**JFK**	0.557	0.682	0.466	
**ROM**	0.729	0.820	0.645	0.577

BID: Bidi; JKF: Jenah Khotifa; ZAR: Jenah Zarzoura; BIS: Biskri; ROM: Rommani; MAH: Mahmoudi.

**Table 5 ijms-20-01352-t005:** Genetic diversity among Tunisian durum wheat landrace populations based on sub-populations number K = 5.

Landraces	*n*	*H* _e_	*F* _ST_
**ZAR**	32	0.27	0.05
**JFK**	25	0.20	0.47
**BID**	34	0.12	0.57
**ROM**	32	0.27	0.30
**BIS+MAH**	73	0.24	0.26

*n*: Number of lines; *H*_e_: Expected heterozygosity (genetic diversity); *F***_ST_**: Measure of genetic differentiation. BID: Bidi; JKF: Jenah Khotifa; ZAR: Jenah Zarzoura; BIS: Biskri; ROM: Rommani; MAH: Mahmoudi.

**Table 6 ijms-20-01352-t006:** List and sample size of the Tunisian durum wheat landraces collected from the center, the south and the Oasis of Tunisia (Set 1).

Landraces	Number of Lines	Province
Bidi	33	Kairouan
Biskri1	31	Gafsa (Djebel ouled ouhiba)
Biskri2	7	Medenine (Zarzis)
Jenah Khotifa1	29	Kairouan
Jenah Khotifa2	3	Tozeur (El Frid)
Jenah Zarzoura	30	Matmata (Oasis of Mareth)
Mahmoudi	30	Gafsa (Snad)
Rommani	33	Gafsa (Djebel Ouled Ouhiba)
Total	196	

**Table 7 ijms-20-01352-t007:** List of the Tunisian durum wheat landraces, Tunisia released varieties, and ICARDA/CIMMYT elite lines (Set 2).

**Tunisian Landraces**	***N***	**Origin**	**Accession Identifier**	**Pedigree**
Agili	1	NGBT	IG 23903	-
Aouadi	1	NGBT	IG 23908	-
Arbi	1	NGBT	IG 23903	-
Azizi	1	NGBT	IG 23904	-
Bayadha	1	NGBT	IG 23905	-
Bidi	3	NGBT	IG 19553; IG 23906; IG 23929	-
Biskri	2	NGBT	IG 19551; IG 23907	-
Chetla	2	NGBT	IG 19555; IG 19557	-
Chili	1	NGBT	IG 23908	-
Derbessi	1	NGBT	IG 23909	-
Hmira	1	NGBT	IG 23910	-
Jneh khotifa	2	NGBT	IG 23915; IG 999	-
Mahmoudi	1	NGBT	IG 23911	-
Richi	1	NGBT	IG 23912	-
Rommani	3	NGBT	IG 19552; IG 19554; IG 19558	-
Sbei	1	NGBT	IG 23913	-
Souri	1	NGBT	IG 23914	-
Swabei algia	1	NGBT	IG 23916	-
Tounsia	1	NGBT	IG 19559	-
Ward bled	1	NGBT	IG 23917	-
**Tunisian Improved Varieties**	***N***	**Origin**	**Accession Identifier**	**Pedigree**
Inrat69 *	1	NGBT	IG 23919	Mahamoudi/Kyperounda
Karim *	1	NGBT	IG 23924	Jori“S”/Anhinga“S”//Flamingo“S”
Khiar *	1	NGBT	IG 23922	Chen/Altar 84
Om Rabii *	1	NGBT	IG 23921	Jori C69/ Hau
Nasr *	1	NGBT	IG 23923	GoVZ512/Cit//Ruff/Fg/3/Pin/Gre//Trob
Maali *	1	NGBT	IG 23920	CMH80A.1060/4/T.TURA/CMH74A.370//CMH77.774/3/YAV79/5/RAZZAK/6/DACK/YEL//KHIAR
**ICARDA/CIMMYT Elite Lines**	***N***	**Origin**	**Accession Identifier**	**Pedigree**
	1	ICARDA	MCHCB-102	OmRabi3/T.*urartu*500651/ch5//980947/3/Otb4//Ossl1/Rfm6
IcaJoudy1 *	1	ICARDA	MCHCB-100	Atlast1/961081//Icasyr1
Nachit *	1	ICARDA	DAWRYT-106	Ameddkul1/T. *dicoccoides* Syrian collection//Loukos
Zeina4 *	1	ICARDA	MCHCB-154	GdoVZ512/Cit//Ruff/Fg/3/Src3
Louiza *	1	ICARDA	--	Rscn39/Til1
Ammar 6 *	1	ICARDA	IDYT37-5	ICAMORTA0472/Ammar7
Ammar 10 *	1	ICARDA	MCHCB-99	Lgt3/Bicrecham1

N: Number of accessions; *: Accessions marked with * are advanced lines; --: Inbred line without identifier; -: Accession without pedigree (Landrace), NGBT: National Gene Bank of Tunisia; ICARDA: International Center for Agricultural Research in Dry Areas.

**Table 8 ijms-20-01352-t008:** Number of durum wheat landraces from the Mediterranean and West Asian countries provided by the ICARDA genebank (Set 3).

Geographical Origin	Number of Landraces
Afghanistan	16
Cyprus	10
Algeria	14
Egypt	12
Ethiopia	11
Greece	18
Iran	12
Iraq	11
Israel	10
Italy	12
Jordan	11
Lebanon	10
Libya	9
Morocco	17
Syria	11
Tunisia	13
Yemen	10

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
