# Peer review of "Genome-Wide Genetic Diversity and Population Structure of Tunisian Durum Wheat Landraces Based on DArTseq Technology"

_ijms, 2019, doi:10.3390/ijms20061352_

Reviewer 1 Report

The manuscript is very clear with results supporting conclusions. Methodology is well applied using novel genotyping techniques.

Results of the work will be of useful interest to breeding programs to increase genetic diversity and to incorporate alleles of interest to the modern varieties.

Author Response

Response to Reviewer 1 Comments and Suggestions

Point 1: The manuscript is very clear with results supporting conclusions. Methodology is well applied using novel genotyping techniques.

Results of the work will be of useful interest to breeding programs to increase genetic diversity and to incorporate alleles of interest to the modern varieties.

Response 1: 

I corrected typos and language mistakes across the document

Reviewer 2 Report

The authors study the genetic diversity of 6 Tunisian durum wheat landraces using DArTseq markers. Actually few studies investigated the Tunisian durum landraces with a large number of molecular markers. Therefore the genotyping of this germplasm may be useful for to detect the genetic diversity. However, the manuscript show different gaps indicating that the manuscript is not yet ready for a scientific publication.

In plant material paragraph the accessions were grouped in three set but in the results the authors shown mainly the data of the set 1. A different number of plants was tested for the three sets of accessions. Therefore the comparisons among the three sets are not reliable.

Not well explained if there is genetic variability within each landrace. Normally in a Principal Component Analysis  (PCA) the first 2 or 3 components explain the 60-70% of the variability. This study includes several components (> 80) to explain the observed molecular variability. The Discriminant Analysis of Principal Components (DAPC) was used to study the genetic structure of Tunisian durum landraces. The optimum number of clusters is obtained with K=6 (line 142). Nevertheless,  in the results is discussed mainly k=5. In addition, to explain the results of each k is excessive and weighs down the understanding. In the Results, the paragraph "Comparison of Tunisian landraces to landraces from Mediterranean and West Asia regions" seems have a marginal role in the manuscript. This paragraph should be deleted or explained extensively.

Little attention was given to the reading of the manuscript. An example is show in the abbreviations (after line 527) in which  DACP and AMOVA indicate Discriminant Analysis of Principal Components. The same error is reported in line 500. The figures appear to be of low resolution and are not legible.

Author Response

Response to Reviewer 2 Comments and Suggestions

Point 1: In plant material paragraph the accessions were grouped in three set but in the results the authors shown mainly the data of the set 1. A different number of plants was tested for the three sets of accessions. Therefore the comparisons among the three sets are not reliable.

Response 1: In this study, we didn’t compare the three sets. We used three different sets of plant material for targeted objectives:

The first set is composed of individual lines from six different landraces collected from the center, the south and the Oasis of Tunisia. The names of the landraces used for this set were attributed based on farmer’s information provided during the collecting missions which could be reliable or not. Moreover, each landrace population could also contain a mixture with other landraces and/or improved varieties. This set allowed to study the genetic diversity between and within the Tunisian landraces and the population structure.

The second set is composed of landraces available within Tunisian genebank, Tunisian improved varieties and elite germplasm from ICARDA and CIMMYT along with the miss-classified lines of the first set. This set is used to study the relationships between different types of germplasm and to shed more light on the miss-classified lines of the first set.

The third set is composed of four lines from each landrace of set 1 and accessions of durum wheat from different Mediterranean and West Asian countries selected at random from the durum wheat collection conserved at ICARDA genebank. This set is used to study the relationships between Tunisian landraces with landraces from other countries and to check if Jenah Zazoura population constitutes a new gene pool.

I made a change in the set 3 in order to clarify that we used only four lines from each landrace and not all the set 1 along with accessions from the Mediterranean and Asian countries as described in details in plant materials (Line 454)

Point 2: Not well explained if there is genetic variability within each landrace.

Response 2: Our results showed that the genetic variability within each landrace is very low as it was described in lines 173 and 174.

“Moreover, the variance between lines within populations decreased from 11.32% for K=2 to reach values close to 0% for K=5.”

Moreover, we used the expected heterozygosity (genetic diversity) in each landrace in order to see which one is the most or less diverse. It was described in the paragraph of the result part “2.4 Genetic diversity and genetic distance between Tunisian landraces”

I added a sentence in the third paragraph of result part “2.3. Population structure of the Tunisian landraces” as a conclusion in order to clarify that the genetic variability within each landrace is very low compared to the genetic variability among landraces:

“Thus, these findings indicated a higher genetic variation among rather than within Tunisian landraces.”

Point 3: Normally in a Principal Component Analysis  (PCA) the first 2 or 3 components explain the 60-70% of the variability. This study includes several components (> 80) to explain the observed molecular variability.

Response 3: The first two or three components from PCA analysis rarely explained a big portion of variance when genotypic data is used. DAPC (Discriminant Analysis of Principal Components) analysis as explained in the materials and methods will use first PCA with higher number of PCs to detect maximum genetic variability, then these PCs are used in a clustering algorithm to detect the number of clusters and last these clusters are used in a discriminate analysis.

The DAPC using the adegenet package for the R software aims to identify and describe genetic clusters. The function of this method displays a graph of cumulated variance explained by the eigenvalues of the PCA. Apart from computational time, there is no reason for keeping a small number of components. We should keep all the information, specifying to retain the maximum number of PCs as described by Jombart, 2015.

Reference

Jombart, T. A tutorial for Discriminant Analysis of Principal Components (DAPC) using adegenet 2.0.0. Available online: http://adegenet.r-forge.r-project.org/files/tutorial-dapc.pdf. (Accessed on 23 June 2015).

Point 4: The Discriminant Analysis of Principal Components (DAPC) was used to study the genetic structure of Tunisian durum landraces. The optimum number of clusters is obtained with K=6 (line 142). Nevertheless, in the results is discussed mainly k=5. In addition, to explain the results of each k is excessive and weighs down the understanding.

Response 4: The optimal number of clusters is obtained with K= 6. However, the number of sub-population K=5 reflect better the differences among landraces as named by farmers. While with K=6 an additional group is added composed of a mixture of lines belonging to different landraces. In order to validate statistically the optimum number of sub-population (K=5), we estimated the significance levels for variance component based on 10000 permutations using randtest function in R-project as described by Excoffier et al. (1992), figure 6 showed in the main document that the number of groups with K=5 was valid and not a result of a random effect.

Reference

Excoffier, L.; Smouse, P.E.; Quattro, J. M. Analysis of Molecular Variance Inferred from Metric Distances among DNA Haplotypes: Application to Human Mitochondrial DNA Restriction Data. Genetics 1992, 131 (2), 479–491.

Point 5: In the Results, the paragraph "Comparison of Tunisian landraces to landraces from Mediterranean and West Asia regions" seems have a marginal role in the manuscript. This paragraph should be deleted or explained extensively.

Response 5: This paragraph is a continuity of the logic in this study. The Jneh Zarzoura landrace was different from both Tunisian landraces and also the advanced germplasm from Tunisia, CIMMYT and ICARDA. The last step was an attempt to check if maybe this landrace was from another neighboring region or not. We have validated that this landrace is not related to any other germplasm from the other regions and hence constitute a different pool within the Tunisian collection. This part allowed also to speculate on the origin of durum wheat landraces.

I added a sentence in the last paragraph of the results part 2.6 as a conclusion to the population structure analysis in order to show the important role of this study. 

The population structure results confirmed that the newly collected landrace Jenah Zarzoura constitutes a new gene pool and showed a genetic similarity between the Tunisian and the North African landraces.”

Point 6: Little attention was given to the reading of the manuscript. An example is show in the abbreviations (after line 527) in which DACP and AMOVA indicate Discriminant Analysis of Principal Components. The same error is reported in line 500. The figures appear to be of low resolution and are not legible.

Response 6: We have read again the manuscript and correct the typos and the language mistakes

I corrected the wrong attribution of AMOVA in the document (In the materials and methods “paragraph 4.4” and in the allocated part for Abbreviations)

I replaced figures 2, 3, 5, 7 and 8 in order to get a resolution of 600 dpi 

Reviewer 3 Report

The manuscript entitled “Genome-wide genetic diversity and population structure of Tunisian durum wheat landraces based on DArTseq technology” is focused on the genetic diversity and population structure analysis of durum wheat. The manuscript falls within the scope and contains useful information for the readers of IJMS. The idea of this manuscript is interesting, most notably because the authors have used Tunisian durum wheat landraces. The analysis is good, and scientific contents are also good. However, the manuscript was not proof read before submission and it contains numerous syntax and grammar errors. I suggest authors to carefully revise the whole manuscript, such as

L74: should be simultaneously

L77: need to revise as tetraploid and hexaploid wheat

L127: Need to rephrase

L172: and the real groups

L322: wrong citation 39-24?

P19: wrong explanation for AMOVA. it stands for Analysis of molecular variance

Author Response

Response to Reviewer 3 Comments and Suggestions

Point 1: L74: should be simultaneously

Response 1: I made change in the line L74 (I eliminated simultaneously as recommended)

Point 2: L77: need to revise as tetraploid and hexaploid wheat

Response 2: I made change in the line L74 (previous sentence “such as in tetraploid wheat and hexaploid wheat” replaced as recommended by “such as tetraploid and hexaploid wheat)

Point 3: L127: Need to rephrase

Response 3: I made change according to the recommendations

Previous paragraph

Cluster analysis using all the lines and based on allele sharing distance (ASD) allowed to identify five clusters and to classify the wrongly assigned lines. The first cluster comprises all the lines of Jenah Zarzoura (Zar). The second cluster contains mainly Bidi lines (Bid). The third cluster could be subdivided into two sub-groups, one of them containing Rommani lines (Rom) and the other is constituted with mixture of lines from other landraces (seven from Biskri (Bis), three from Jenah Khotifa (jkf) and two from Bidi (Bid)). Cluster 4 had the majority of Jenah Khotifa lines (jkf) but contains also lines of Rommani (4), Biskri (2) and Mahmoudi (1). The last cluster constitutes the largest one and gathers almost all lines of Biskri (Bis) and Mahmoudi (Mah) (Figure 3).

Correction according to the recommendations

Cluster analysis of the 196 lines derived from the six Tunisian landraces was performed using the allele sharing distance (ASD) method. Results from cluster analysis allowed to identify five clusters and the lines wrongly assigned to some landraces. The first cluster comprises all the lines of Jenah Zarzoura (Zar). The second cluster contains mainly Bidi lines (Bid). The third cluster could be subdivided into two sub-groups, one containing Rommani lines (Rom) and the other having a mixture of lines from other landraces (seven from Biskri (Bis), three from Jenah Khotifa (jkf) and two from Bidi (Bid)). Cluster 4 had the majority of Jenah Khotifa lines (jkf) but contains also four lines of Rommani, two of Biskri and one of Mahmoudi. The last cluster, the largest one gathers almost all lines of Biskri (Bis) and Mahmoudi (Mah) (Figure 3).

Point 4: L172: and the real groups

Response 4: I replaced real grouping by real groups in the entire document

Point 5: L322: wrong citation 39-24?

Response 5: The reference 39 was replaced by 49 and all the following references have been inserted and adjusted respecting the order of the references.

Point6: P19: wrong explanation for AMOVA. it stands for Analysis of molecular variance

Response 6: We have read again the manuscript and correct the typos and the language mistakes

We corrected the wrong attribution of AMOVA in the document (In the materials and methods “paragraph 4.4” and in the allocated part for Abbreviations).

NB: You will find here under the revised version of the manuscript according to the suggestions 
